# Octenyl Succinic Acid Starch-Stabilized Vanilla Essential Oil Pickering Emulsion: Preparation, Characterization, Antioxidant Activity, and Storage Stability

**DOI:** 10.3390/foods11070987

**Published:** 2022-03-29

**Authors:** Yitong Wang, Bo Li, Libin Zhu, Ping Wang, Fei Xu, Yanjun Zhang

**Affiliations:** 1Spice and Beverage Research Institute, Chinese Academy of Tropical Agricultural Sciences, Wanning 571533, China; wangyitong9705@163.com (Y.W.); 13654529477@163.com (B.L.); xufei_0302054@163.com (F.X.); 2School of Forest, Northeast Forestry University, Harbing 150040, China; wangping2178@nefu.edu.cn; 3Key Laboratory of Processing Suitability and Quality Control of the Special Tropical Crops of Hainan Province, Wanning 571533, China; 4College of Light Industry and Food Engineering, Guangxi University, Nanning 530003, China; 5College of Food Science, Heilongjiang Bayi Agricultural University, Daqing 163319, China; zlbzs196179@163.com

**Keywords:** vanilla essential oil, OSA—starch, pickering emulsion

## Abstract

Applications for vanilla essential oil extracted from vanilla pods have been limited since the effective components of vanilla could be easily influenced by environmental factors, such as temperature, light, and oxygen, which hinder their effectiveness. In this study, vanilla essential oil was encapsulated in a Pickering emulsion with octenyl succinic acid starch (OSA—starch). The optimal process conditions for emulsion preparation were determined as 5% vanilla essential oil phase with 2.5% OSA—starch when they were ultrusonicated for 3 min at 470 W. Under these conditions, the minimum particle size was 0.456 μm, the oil droplets were completely encased by starch, and no new chemical bonds were formed. The smallest particle size was produced at a pH of 4 and 500 mM ion concentration. The antioxidant activity of the emulsion was greater than that of the pure vanilla oil at the same oil content. After 24 h storage, the antioxidant activity of the emulsion was enhanced, and the vanilla essential oil was slowly released in the emulsion. These results indicated that the vanilla essential oil encapsulated in a Pickering emulsion with octenyl succinic acid starch showed its tremendous potential for use in the food industry.

## 1. Introduction

Known as “Flavor Queen.”, vanilla (*Vanilla planifolia Andrew*) is a valuable natural flavoring agent that is grown in tropical and subtropical areas. Vanilla is the only orchid with edible fruits and is commercially cultivated for its pods [1]. Most studies on vanilla have focused on its volatile compounds, which contribute to its complex aroma. Over 300 compounds are responsible for the flavor profile of vanilla [2]. As a characteristic compound of vanilla, vanillin has antibacterial, anti-inflammatory, and antimutagenic properties at a concentration of 1–3% (dry weight) of fermented vanilla pods [3]. Although its yields are low, vanilla essential oil is widely used in perfumes, body oils, and other health and beauty products [4]. However, the principal components of vanilla oil are volatile and unstable in the presence of air, light, humidity, high-temperature environments, and its fragrance is not sustainable, which limits the value. Therefore, reducing the volatility of vanilla essential oil, improving its stability, and thereby increasing its value, has become an enormous challenge.

Essential oils are the essence of plants and spices, which are fragrant and distributed throughout different plant organs, such as roots, stems, leaves, flowers, fruits, and seeds [5]. Essential oils are extracted through different techniques, including steam distillation and organic solvent extraction, supercritical CO_2_ extraction, ultrasonic-assisted extraction, ultrasonic extraction, and molecular distillation. Compared to extraction methods, such as distillation by steam and organic solvent extraction, supercritical CO_2_ extraction has many advantages, such as a low working temperature for reducing energy consumption, saving heat-sensitive compounds while avoiding residual toxic solvents, and retaining the sensory characteristics of the natural plant. Donelian et al. [6] compared the yields of patchouli essential oil by supercritical CO_2_ and steam distillation; the results showed that the supercritical CO_2_ technique at 14 MPa and 40 °C gave the better yield (5.07%), which was higher than that for steam distillation (1.50%). Moon et al. [7] compared the ethanol extraction method (0.9% yield) and supercritical CO_2_ extraction (2.8% yield), on the production of the essential oil from *A. heterotropoides.* In addition, the fatty acid content was higher, and the linoleic acid content was increased from 21.4% to 26.4% using the supercritical CO_2_ method.

As the essential oil is a water-insoluble compound, many strategies have been developed to improve its water dispersibility, such as incorporating them into carriers, emulsifiers, and solvents within the formulations. These products were produced by microcapsules, nano-emulsions, Pickering emulsions (PE), liposomes, and complexes [8]. Pickering emulsions have attracted increasing attention because of their core-shell structure, where the essential oil can be encapsulated within the inner core [9]. Compared with nano-emulsions, the use of toxic reagents is reduced, making the prepared emulsion more environmentally friendly and safe when the formulation is a Pickering emulsion. In addition, Pickering emulsions are relatively stable emulsions produced using solid particles to stabilize the interface between two incompatible liquids, reducing the interfacial energy of the system [10]. Pickering emulsions are widely used in food, drug, and functional substance delivery systems because of their advantages, including high environmental compatibility, stability, controlled release, anti-aggregation, and reduction of lipid oxidation.

Until now, no study has focused on the evaluation of the properties and activity of Pickering emulsion with vanilla essential oil. In this study, the vanilla essential oil of Pickering emulsion has been prepared to investigate its characteristics, including the volatile constituents, fatty acid constituents, grease characteristic value of vanilla essential oil, particle size, and stability. In addition, the antioxidant properties of vanilla essential oil and their Pickering emulsions were also determined. This study aims to provide a method of preparing Pickering emulsion with vanilla essential oil. The prepared emulsion can be used to improve the instability of vanilla essential oil, and at the same time enable it to possess slow-release properties and improve its antioxidant properties. These results could help vanilla essential oil become better utilized for different fields.

## 2. Materials and Methods

### 2.1. Materials

Octenyl succinic anhydride-modified starch (OSA—modified starch) was purchased from Shanghai yuan ye Bio-Technology (Shanghai, China). Vanilla pods were harvested in November 2019 from the garden at the Spice and Beverage Research Institute (Hainan, China). 2,2-diphenyl-1-picrylhydrazyl (DPPH), 2,2-Azino-bis (3-ethylbenzothiazoline-6-sulphonic acid) diammonium salt (ABTS), and deionized water produced by a Milli-Q system (Millipore, Bedford, MA, USA) were used in all experiments.

### 2.2. Extraction of Vanilla Essential Oil

The supercritical CO_2_ extraction method was used to extract vanilla essential oil [11]. First, the sample of fermented vanilla pods was dehydrated in an electric blast drying oven at 50 ± 5.0 °C for 30 h to eliminate a moisture water content of 6%. The size of vanilla pods was about 7 cm. The pods were cut to 2–3 cm and then placed in a high-speed disintegrator (Herbal Crusher YF-1000). The size of crushed pod powders was about 200 μm. Then, vanilla powder (500 g) was placed in an extraction kettle. The analytical conditions employed were as follows: heating temperature 35 °C, separation heating temperature of 36 °C, CO_2_ flow rate of 68.5 L/h, extraction kettle pressure, separation kettle I pressure, and separation kettle II pressure of 29, 10, and 4 MPa. The extracted vanilla essential oil was used in subsequent studies. 

### 2.3. Gas Chromatography–Mass Spectrometry Conditions for Vanilla Essential Oil and Fatty Acid

For component analysis, GC–MS analysis of vanilla essential oil was performed on a THERMO ISQ equipped with a J&W DB-5 capillary column (0.25 mm × 30 m × 0.25 μm) [6]. The vanilla essential oil was diluted 50 times with n-hexane, passed through a 0.45 μm microporous membrane, and injected. The operating conditions were as follows: the GC temperature program was first held at 50 °C, increased at 3 °C/min to 75 °C, increased at 1.5 °C/min to 140 °C, then increased at 10°C/min to 230 °C, and held at 230 °C for 2 min; finally, the temperature was increased to 280 °C at 20 °C/min and maintained for 3 min. The inlet temperature was 250 °C, the flow rate was 1.0 mL/min. The sample injection volume was 1 μL with no split. Mass spectrometry conditions were as follows: EI was used as the ion source, the ion source temperature was 230 °C, the electron energy was 70 eV, and the sector mass analyzer was set to scan from 30 to 450 amu. The NIST library search results and references were used for qualitative analysis, and the relative content was calculated using the peak area normalization method.

### 2.4. Measurements for Acid Value (AV), Peroxide Value (PV), Iodine Value (IV), and Saponification Number (SN)

The measurement methods for acid value, peroxide value, iodine value, and the saponification number were based on the AOAC standard [12].

### 2.5. Preparation of Pickering Emulsion

Pickering emulsions were prepared as reported Xie [13] with some modifications, as shown in Figure 1. Pickering emulsions containing 5, 7.5, 10, 12.5, and 15% (*v*/*v* of total emulsion) vanilla essential oil with different ratios of OSA—starch (0.5, 1.0, 1.5, 2.0, 2.5, and 3.0% of total emulsion) were prepared. The starch solution was heated to 85 °C and then lowered to 50 °C. Subsequently, vanilla essential oil was added to the starch solution, stirred in a high-shear machine at 10,000 rpm for 3 min. An ultrasonic cell disruptor was then used to sonicate samples of this mixture for 1, 2, 3, 4, 5, and 6 min at 50, 150, 250, 350, 450, and 550 W, respectively. The temperature was maintained at 25 °C ± 3 °C during ultrasonication This method was used to develop the optimal formulation of emulsions, depending on the concentration of OSA—starch, ultrasonic power, and ultrasonic time. These independent variables were studied at three levels (−1, 0, 1), and the central point of the design (0, 0) was carried out in triplicate to determine the reproducibility of the emulsion process. (Table 1). The volumetric mean droplet diameter (D_4,3_) was used as a dependent variable to optimize the emulsion formulation. A total of 10 mL of the emulsion was produced for the next experiment.

### 2.6. Particle Size Distribution of Pickering Emulsion

Mastersizer 3000 was used to measure the mean sizes and particle size distribution of the Pickering emulsion at 25 ± 1 °C. The refractive indices were 1.46 and 1.33, respectively, 300 s sample equilibration time was allowed, each emulsion sample was diluted nine times with an interval of 10 s prior to analysis [14].

### 2.7. Confocal Laser Scanning Microscopy (CLSM)

The PE was also observed using confocal laser scanning microscopy (CLSM). Nile red dye was used to stain the oil phase, and Nile blue was applied to stain the OSA—starch. The tested PE (2 mL was dyed with a mixed fluorescent dye solution consisting of 50 μL 1 mg/mL Nile red and 50 μL 1 mg/mL Nile blue for 5 min, stored in the dark for 2 h, and then measured. The fluorescent dyes were excited by either an argon laser at 488 nm for Nile red or a helium-neon (He-Ne) laser at 633 nm for Nile blue [9]. 

### 2.8. Fourier Transform Infrared Spectra (FTIR)

FTIR spectra of the OSA—starch, vanilla essential oil, and PE were obtained using a Fourier near-infrared spectrometer, recorded using KBr pellets for solid samples. First, 200 mg of KBr was ground into a homogeneous powder. A transparent slice was created under a pressure of 10 MPa. The sample was kept for 60 s in a rapid infrared dryer. The same volume of vanilla essential oil, starch solution, and Pickering emulsion was applied to the surface and placed in the beam of the instrument. The acquisition conditions were as follows: spectral width of 4000–400 cm^−1^, 64 accumulations, and 4 cm^−1^ resolution.

### 2.9. Emulsion Stability

Freshly prepared Pickering emulsion containing 2.5% OSA—starch and 5% vanilla essential oil was subjected to a series of environmental stresses, including a range of pH and ionic strength conditions. The pH and ionic strength were set as follows: the PEs were placed in 20 mL beakers, adjusted to investigated pH values (2–8) using NaOH or HCl solutions or salt level (100–500 mM) by adding NaCl, and then transferred into glass tubes.

### 2.10. Determination of Antioxidant Properties

DPPH radical scavenging activity was measured by slightly modifying a previously reported method [10]. Samples (2 mL) with different concentrations were added to 2 mL of DPPH-ethanol solution (0.2 mm) and immediately stirred. The mixed solution was stored at room temperature for 30 min in the dark, and the absorbance at 517 nm (*A_t_*) was recorded using a UV spectrophotometer. The scavenging activity (%) was calculated using Equation (1):(1)$$\mathrm{DPPH}\;\mathrm{scavenging}\;\mathrm{activity}\;\left(\%\right)=\left(1-\frac{\left(A_{i}-A_{j}\right)}{A_{k}}\right)\times 100$$
where *A_i_* is the absorbance of DPPH-ethanol, *A_j_* is the absorbance of PE, DPPH-ethanol is replaced by ethanol, *A_k_* is the absorbance of DPPH-ethanol, and PE is replaced with a starch solution of the same concentration.

The working solution was obtained by mixing ABTS (7.4 mmol/L) with potassium persulfate (2.6 mmol/L) and storing at room temperature for 12–16 h. On the day of analysis, the working solution was diluted with ethanol to an absorbance of 0.70 ± 0.03 nm. A 100 μL aliquot of different concentrations was added to the working solution, the absorbance of which was measured at 734 nm after 6 min. Ethanol was used as a blank, and the free radical scavenging activity was calculated using Equation (2):(2)$$\mathrm{ABTS}\;\mathrm{scavenging}\;\mathrm{activity}\;\left(\%\right)=\frac{\left(A_{i}-A_{k}\right)}{A_{i}}\times 100$$
where *A_i_* and *A_k_* are the is the absorbances of blank and PE, respectively.

### 2.11. Experimental Design Diagram

The overall idea of the test and process were shown in Figure 2.

### 2.12. Statistical Analysis 

All experiments were performed in triplicate, and the results were recorded as means and standard deviations. Statistical analysis was performed using the SPSS 13.0 statistical analysis program. Data were subjected to analysis of variance (ANOVA), and Duncan’s multiple range tests were used to establish the significance of differences (*p* < 0.05) among the mean values.

## 3. Results and Discussion

### 3.1. GC–MS Analysis

#### 3.1.1. Volatile Constituent Analysis of Vanilla Essential Oil

The composition analysis of the vanilla of GC–MS is shown in Figure 3, and the corresponding compound names are listed in Table 2.

The volatile components of vanilla analyzed by GC–MS have been mainly focused on organic solvent extraction, while the composition of vanilla essential oil extracted by a supercritical fluid has not been reported. The GC—MS qualitative data showed the retention time (RT) of 6.30–7.36, and 27 volatile components in which the aldehydes accounted for 47.00%, acids accounted for 19.61%, and alcohols accounted for 16.01%. The main substances in vanilla essential oil were 4-methoxy-benzenemethanol (RC: 8.69%), vanillin (RC: 30.54%), and n-hexadecanoic acid (RC: 4.83%), according to the retention time (RT, 28.25 min; RT, 37.53 min; and RT, 60.30 min). A different observation was reported by Pérez-Silva et al. [1], who detected 65 volatile compounds using organic solvent extraction. The differences could be explained by that, the supercritical extraction of vanilla essential oil has fewer volatile components compared with the organic solvent extraction. A similar result was also observed of Yamini et al. [11], who reported that the composition of the *Salvia mirzayanii* essential oil extracted by supercritical CO_2_ (20 compounds) was lower than that of the steam method (34 compounds). Based on Yamini et al. [11], the number of compounds detected in supercritical extraction essential oils was lower, but the yield was greater than the other methods, such as water steam extraction (41.5% as compared to 7.6%). 

In this study, vanillin (30.54%) was the main component of vanilla, which was slightly higher than the result found by Vega et al. [2]; vanillin content ranged from 19–30 wt% by changing the extraction parameters. 

Besides the main substances, the overall aroma of vanilla oil was also affected by trace components of volatiles based on the early results [5]. In this study, the contents of phenethyl alcohol and 4-methoxybenzaldehyde were 0.23 ± 0.14% and 0.14 ± 0.10%, respectively. Similar results were obtained by Dong et al. [4] on the GC-MS analysis of vanilla extracted with dichloromethane, that the phenethyl alcohol and 4-methylbenzaldehyde in Hainan vanilla were 0.095% and 0.07%, respectively. 

#### 3.1.2. Fatty Acid Constituent Analysis of Vanilla Essential Oil 

The fatty acid content of the vanilla essential oil was determined by GC-MS, as shown in Table 3 and Figure 4.

The fatty acids of vanilla essential oil revealed the presence of six compounds, including myristic acid (C14:0) (0.07%), palmitic acid (C16:0) (19.91%), stearic acid (C18:0) (7.88%), oleic acid (C18:1) (13.32%), linoleic acid (C18:2) (58.07%), and linolenic acid (C18:3) (0.74%). The saturated fatty acid content was 27.79%, of which the highest content was palmitic acid. The unsaturated fatty acid content was 72.11%, and the highest content was linoleic acid. The results are similar to those of soybean oil reported by Chandrashekar et al. [15], who indicated that 84% of unsaturated fatty acids in soybean oil and linoleic acid (54%) was the main component. Compared with soybean oil, vanilla essential oil contains the higher amounts of palmitic acid and stearic acid. Similarly, Chen et al. [16] reported that the unsaturated fatty acid content of almond oil extracted by supercritical CO_2_ was approximately 78%. 

Mohammed et al. [17] reported vanilla essential oil to show lower saturated fatty acid content than that of coconut oil, which contains more than 90% saturated fatty acids, as well as higher myristic acid (19.58%) and a lower content of stearic acid (3.16%) than those of vanilla essential oil. Compared to vanilla essential oil, almond oil had a higher content of oleic acid (48%) and linoleic acid (26%). Ariffin et al. [18] found that the fatty acid content of vanilla essential oil was similar to that of dragon fruit seed oil (23.59%), and similar contents of palmitic acid (17–18%) and stearic acid (approximately 5%).

### 3.2. Physicochemical Properties of Vanilla Essential Oil

The acidity, iodine, peroxide, and saponification values are the major characterization parameters for oil quality. The characteristic values for vanilla essential oil are shown in Table 4. 

Although the main fatty acid content of vanilla essential oil was similar to that previously reported for soybean oil, the iodine value of soybean oil was higher (ranging from 116.9–118.1 mg/100 g oil) based on the results reported by Abdulkarim et al. [19]. The peroxide value of soybean oil increased rapidly over time because of the high linolenic acid content (5.55%). In addition, the saponification value of vanilla essential oil was lower than coconut oil (262.55 mg KOH/100 g oil), and the iodine value was similar to coconut oil (6.05 g I_2_/100 g oil) based on Mohammed et al. [17], due to the fact that vanilla essential oil has slightly more unsaturated fatty acids than coconut oil.

### 3.3. Particle Size Distribution of Pickering Emulsion

Figure 5 shows the particle size change and particle size normal distribution diagram of Pickering emulsions when different vanilla essential oil contents (5%, 7.5%, 10%, 12.5%, and 15%) were used as a variable under the condition of fixed OSA—starch concentration (2.5%).

The particle size of the emulsion gradually increased with an increase in the oil phase concentration, and the significant difference gradually became more extensive among the particle diameters, causing the emulsion to become unstable. When the vanilla essential oil concentration was 5%, the minimum of the five different oil-phase concentration emulsions was 0.485 μm (D_4,3_). According to Jia et al. [20], the lower oil phase concentration caused high-density starch particles to be better enclose the surface of the oil droplets. The three-dimensional network structure or layer-by-layer enclosed droplets contributed to the stabilization of the high interfacial surface area during the limited coalescence process. As researched by Liu et al. [21], with the content of vanilla essential oil increasing in the Pickering emulsion, the same concentration of OSA—starch gradually failed to form a multilayer structure, which caused a single layer of OSA—starch granules to stabilize the vanilla essential oil, resulting in a gradual increase in the particle size. When the vanilla essential oil concentration was 15%, the particle size increased to 1.28 μm. Correspondingly, the bimodal peaks were exhibited in particle size distribution graph at this time, indicating that the emulsion contained both small and relatively large droplets. As a result, the emulsion was prepared with a vanilla oil concentration of 5% to obtain smaller emulsion droplets in the next experiment.

The effect of the oil phase concentration far exceeds the other three factors on particle size (ultrasonic time, ultrasonic power, and OSA—starch concentration). Therefore, the other three factors were optimized on the particle size of emulsion stability when the response surface method was used in the following experiments.

### 3.4. Interpretation of the Response Surface Method

The influence of the ultrasonic time on the particle size of the Pickering emulsion showed a first decreasing and then increasing trend. As shown in Figure 6 (left 1), prolonging the ultrasonic time was beneficial for fully dispersing the droplets and improving the emulsion stability. When the ultrasonic time was increased from 1 min to 3 min, the particle size decreased from 0.87 to 0.63 μm. Based on the results of Low et al. [22], the decrease in the emulsion particle size was likely due to the gradual decrease in the viscosity of the emulsion under ultrasonication. With the increase in ultrasonic time, the temperature increased and the viscosity decreased. Therefore, the lower viscosity makes the cavitation effect stronger, and the droplets smaller. When the ultrasonic time was longer than 3 min, the particle size was gradually increased, and the particle size was 0.76 μm when the ultrasonic time was 6 min, indicating that the emulsion stability decreased with the extension of the ultrasonic time.

The effect of the ultrasonic power on the emulsion showed a first decreasing trend and then increasing trend for the particle size. As shown in Figure 6 (left 2), when the ultrasonic power increased from 50 W to 150 W, the particle size decreased sharply. As the ultrasonic power increased to 450 W, the minimum size of 0.51 μm was found. According to Tang et al. [23], the increase in ultrasonic power increased the number of bubbles due to ultrasonic cavitation, and the energy around the bubbles facilitated their dispersion, leading to a decreased trend for the particle size. The particle size of the emulsion increased with the further increase in the ultrasonic power. This may be ascribed to the temperature gradually increased during the ultrasonic process, and the molecular motion speed increased, resulting in aggregation. Figure 6 (left 3) shows D_4, 3_ vanilla essential oil Pickering emulsions prepared with different OSA—starch concentrations (0.5–3.0%). The emulsion particle size decreased gradually, with increased OSA—starch concentration. As stated by a previous report [13], the droplet size of the emulsion decreased when the starch concentration was increased, until it reached a constant value for the droplets. At this point, the steric hindrance increased, and the droplet aggregation rate decreased. When the OSA—starch concentration was 2.5%, the minimum emulsion particle size was 0.48 μm. When the OSA—starch concentration was increased to 3%, the emulsion particle size increased, the reason for this may be the increased starch particle coverage on the droplets. 

Figure 6 (right) shows a diagram of the average droplet size D_4,3_ relative to two factors when one of the three remained constant. A minimum value of 0.46 μm was observed in the experimental area. In Figure 6 (right 1), when the concentration of OSA—starch was 2.5% and the ultrasonic time was increased from 1 min to 3 min, the droplet size gradually decreased to the lowest point at an ultrasonic power of 450 W. With an increase in ultrasonic time and ultrasonic power, the droplet size increased. When the ultrasonic power and ultrasound time exceeded 450 W and 3 min, respectively, the particle size increased significantly (*p* < 0.05, right 2). This may be due to the rapid increase in temperature during the ultrasonic emulsification process, especially when the ultrasonic power was high. The particles were aggregated in the emulsion, resulting in an increase in the emulsion size. At an ultrasonic time of 3 min, ultrasonic power of 450 W, and an OSA—starch concentration of 2.5%, the droplet size was the smallest 0.47 μm, as shown in Figure 6 (right 3).

### 3.5. Optimization of Vanilla Essential Oil Process by Response Surface Methodology

The regression equation model can be established by statistical analysis of the 17 groups of experimental data, as shown in Table 5.

$Y=0.46-0.047A-0.015B-0.018C-0.014AB+0.00375AC-0.0025BC+0.092A^{2}+0.057B^{2}+0.043C^{2}$. The quadratic equation model was significant (*p* < 0.05) and R_2_^Adj^ = 0.9484, indicating that the experimental design was reliable and that the fitted quadratic regression equation was suitable. The particle size of the Pickering emulsion could be predicted using this model under different preparation conditions. The minimum theoretical value of the Pickering emulsion was 0.46 μm. The optimal process conditions were ultrasonic time of 3.12 min, ultrasonic power of 479.03 W, and OSA—starch concentration of 2.61%. To verify the best emulsification process, the optimal conditions were adjusted as starch concentration of 3%, ultrasonic power of 450 W, and ultrasonic time of 3 min. The verification test was carried out, and the particle size of the emulsion was 0.47 ± 0.01 μm, which was close to the predicted value of 0.46 μm, indicating that the vanilla essential oil emulsion process model was optimized by the response surface method.

### 3.6. Emulsion Morphology of Pickering Emulsion

The distribution of OSA—starch and vanilla oil in the Pickering emulsion was visualized using the fluorescent dyes, Nile blue (green color) and Nile red (red color), respectively. The CLSM clearly showed that there was a green halo around the oil droplets, as shown in Figure 7. When the 5% vanilla essential oil Pickering emulsion was observed at 600×, green particles surrounded by red droplets were observed, and the diameter of the emulsion particles was small. This result reveals that the Pickering emulsion was stabilized by the OSA—starch particles at the interface between the oil and water phases, which could be used as a granular emulsifier, and the amphiphilic nature was exhibited for modified starch.

### 3.7. FTIR

As shown in Figure 8, the Fourier infrared spectroscopy can be used to further illuminate the interaction between OSA—starch and vanilla essential oil in Pickering emulsion from the perspective of the molecular structure of the substance and the chemical bonds existing in the substance. The entire range of 400–4000 cm^-1^ was covered by the absorption peak of the sample, in which the PE showed the same as the infrared spectrum of OSA—starch and a significant difference from the infrared absorption peak of vanilla essential oil. 

According to previous report [24], the broad peak around 3448.13 cm^−1^ indicated the presence of hydroxyl (-OH). The 1638.61 cm^−1^ peak was the adsorbed water bending vibration in the OSA−starch. As described by Simsek et al. [25], the peak at 1023.37 cm^−1^ was attributed to the C-O stretching of the C-O-C in the polysaccharide, while the peaks at 1078.89 and 1153 cm^−1^ are characteristic of the anhydroglucose ring C−O stretching. The absorption peak of the unsaturated functional group at 2331.33 cm^−1^ in vanilla essential oil and the skeletal stretching vibration C−C absorption peak of the aromatic ring at 1513.96 cm^−1^ [26] disappeared during the preparation of Pickering emulsion. The infrared spectra of Pickering emulsion and vanilla essential oil showed a significant difference in peak shape between 1500–1700 cm^−1^ and 2300–2800 cm^−1^. No new peak shapes or positions were observed when the infrared spectra of vanilla essential oil, OSA—starch, and the Pickering emulsion were compared. This further confirmed that the formation of the Pickering emulsion was due to the electrostatic interaction between OSA—starch and vanilla essential oil rather than a chemical reaction.

### 3.8. Emulsion Stability

#### 3.8.1. The Influence of pH on Emulsion Particle Size

As shown in Figure 9, the particle size increased and then decreased as the pH increased. At low pH (2–4), the particle size increased from 0.49 to 0.53 μm, and the size of the Pickering emulsion droplets remained unchanged and relatively small. According to Li et al. [27], this could be due to the difficulty of the particles to aggregate, caused by a combination of strong electrostatic and steric repulsion. The significant increase (*p* < 0.05) in the particle size was observed as the pH value increased (5–7). According to Fang et al. [14], this could be because the change in the pH of the system led to changes in the surface charge and interface structure of the particle. The degree of close connection between the emulsions finally affected the stability of the emulsion, which was manifested as an increase in the particle size of the emulsion. 

When the pH was approximately 4 (acidic), the static charge between the OSA—starch particles with negative charges increased significantly (*p* < 0.05), and the droplets did not aggregate quickly. Although the particle size increased slightly, this change was not significant (*p* > 0.05). This was different from the report by Zhang et al. [28], where fish oil microcapsules were decomposed under acidic conditions and released the core material, resulting in forming a new interface based on the broken OSA—starch. When the pH was 8, the emulsion particle size decreased. This was similar to the result reported by Boostani et al. [9], that the particle size decreased from 91.4 μm to 47.7 μm, with pH 7 to 8. This phenomenon occurs because the electrostatic repulsion increases between the droplets, and the balance between the droplets can be maintained without agglomeration, thereby reducing the particle size. 

#### 3.8.2. Effect of Salt Concentration on Emulsion Particle Size

As shown in Figure 10, the emulsion particle size increased and then decreased with an increase in the NaCl concentration. According to Li et al. [27], there are two possible reasons for this phenomenon: (1) At low ion concentrations (0–200 mM), the repulsion intensity was smaller than the interaction force between ions. After the addition of NaCl, an electrostatic shielding effect was generated, and the electrostatic repulsion and spatial steric hindrance between droplets could be decreased, resulting in poor internal uniformity. (2) The decrease in droplet size of the emulsion caused by the increase in NaCl concentration was still caused by the electrostatic shielding effect. According to Boostani et al. [9], the emulsion being more stable at high concentrations could be due to the larger emulsion droplet size being more prone to improving the stability of the encapsulated oil. 

#### 3.8.3. Effect of Storage Time on Emulsion Particle Size

The emulsions obtained under optimal conditions were held at 25 ± 0.5 °C for 14 days to study the storage stability of the Pickering emulsions. During different time spans (1, 3, 5, 7, and 14 days), the particle size and distribution of the Pickering emulsion were measured as Figure 11, the appearance of the emulsion system did not appear to be stratified. According to Wu et al. [8], when the particle size and distribution of the emulsion system did not change significantly after a period of observation, it was considered stable. 

It can be seen from the Figure 11 that with the increase in storage time, the emulsion showed a gradually increasing trend of D_4,3_. After seven days, the particle size of the emulsion was not increased significantly. The particle size increased slightly with an increase in storage time due to the close arrangement of solid particles on the outer surface of the oil droplets. The particle size of the emulsion increased by about 0.5 μm in 7–14 days. According to Biduski et al. [29], the aggregation of small droplets, which was formed into large droplets, indicated the instability of the system. The process is accelerated by the Ostwald ripening process, which gradually gathers and increases to create larger droplets. Emadzadeh et al. [30] published similar results for garlic oil. The particle size was 1.0 μm when the garlic essential oil microcapsules were prepared using β-cyclodextrin. After 60 days storage, it increased to 2.7 μm because the wall material was dissolved during storage and the oil droplets were aggregated, increasing the particle size.

### 3.9. Determination of Antioxidant Properties

As shown in Figure 12 (left), after OSA—starch stabilized the preparation of the Pickering emulsion, the oxidation resistance of the emulsion was lower than that of pure oil after 30 min. Moreover, the DPPH**·** scavenging ability of the Pickering emulsion was improved after 24 h, indicating that the scavenging ability was higher than that of the Pickering emulsion at 30 min, also explaining that Pickering emulsion was released slowly. This phenomenon became more evident as the oil concentration increased. The present study was similar to Shah et al. [31], in which the antioxidant, curcumin, was added to corn oil, and chitosan was used as the solid particle to prepare a Pickering emulsion. The free radical scavenging rates were 15.96% for curcumin and 49.58% for the Pickering emulsion. Estrada et al. [32] also reported that a good linear relationship was found between embedding vanillin with whey protein as the emulsifying material and DPPH· scavenging ability. When the same amount of pure vanillin was used in microcapsules, its EC_50_ was twice that of pure vanillin (17.2 mg/mL < 36.2 mg/mL). 

The ABTS ^**+**^ scavenging activities of the vanilla essential oil and Pickering emulsion were increased with increasing concentration (Figure 12 (right)). The ABTS ^**+**^ scavenging ability of the Pickering emulsion at different concentrations was greater than that of pure vanilla essential oil. The significant differences in the ABTS·**^+^**scavenging ability were observed between pure and emulsified vanilla essential oil after 6 min (*p* < 0.05). The free radical scavenging ability of Pickering emulsion was significantly increased after 24 h compared to 6 min. The reason may be the amphiphilic OSA—starch, and the emulsion droplets became smaller after the vanilla essential oil was prepared as Pickering emulsion. This led to the increased specific surface area and increased the contact area with ABTS ^**+**^. The starch shell was gradually broken and the solubility of vanilla essential oil was slowly increased after the preparation of Pickering emulsion, which was also one of the reasons for the high free radical scavenging rate of the emulsions.

## 4. Conclusions

Vanilla essential oil is commonly used as a perfume product, and no research has focused on the stability and functionality of its Pickering emulsion. In this study, the OSA—starch was used as solid particle emulsifier to stabilize vanilla essential oil Pickering emulsion. Six main fatty acids and 27 kinds of volatile components were detected in vanilla essential oil. Compared to other essential oils, the advantages of vanilla oil show a typical flavor of vanillin ingredient, and a higher level of linoleic acid content, which is beneficial to the human body. The Pickering emulsion was optimized at 5% of vanilla essential oil, 2.5% of OSA—starch, 3 min, and 470 W of ultrasonication, in which the minimum particle size of the Pickering emulsion was 0.46 μm. The oil droplets were completely wrapped, and the interface was relatively smooth by the use of a laser confocal microscope. No new chemical bonds were formed between the oil and starch by Fourier infrared spectroscopy. The particle size of the Pickering emulsion was the smallest at pH4 and the smallest particle size (0.47 μm) was found for 500 mM at different salt ion concentrations. With the storage time increasing, the particle size of the Pickering emulsion was gradually increased. The particle size did not change in the 7 days until the size was increased to 0.50 μm from 7 days to 14 days. Furthermore, Pickering emulsions showed the similar antioxidant potential with vanilla essential oil. The results could be used as reference to design stable formulations with Pickering emulsion, prompting more utilizations and investigations on the wider applications for vanilla essential oil in the food industry. This research provides a green and environmentally friendly way for the storage of essential oil and a proposed theoretical basis for the future application and development of vanilla essential oil.

## Figures and Tables

**Figure 1 foods-11-00987-f001:**
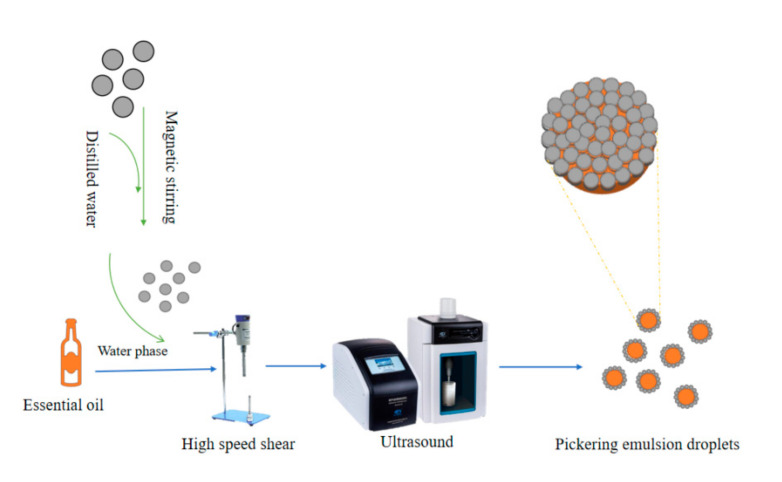
Schematic diagram of Pickering emulsion prepared by shear and ultrasound.

**Figure 2 foods-11-00987-f002:**
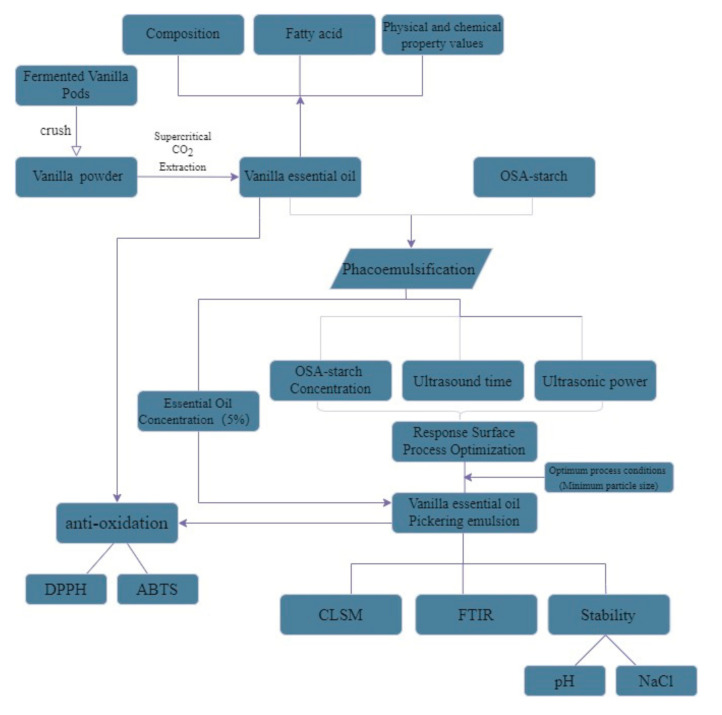
Experimental design roadmap.

**Figure 3 foods-11-00987-f003:**
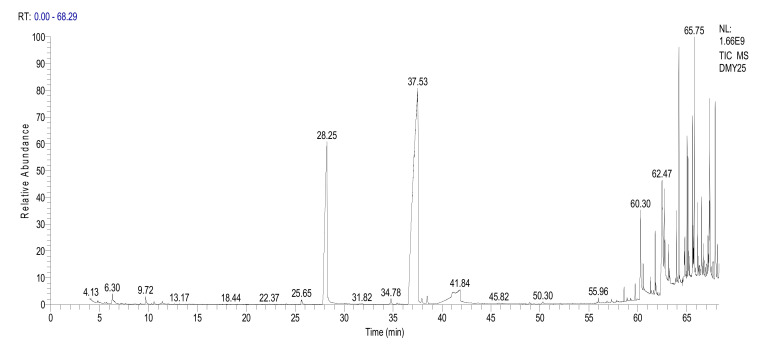
GC–MS qualitative data of value vanilla essential oil.

**Figure 4 foods-11-00987-f004:**
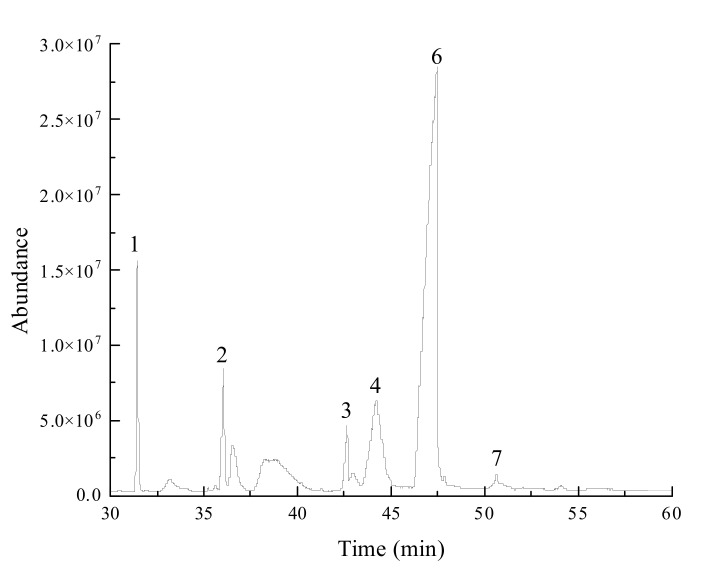
GC–MS total ion chromatogram of fatty acid chemical composition in vanilla oil.

**Figure 5 foods-11-00987-f005:**
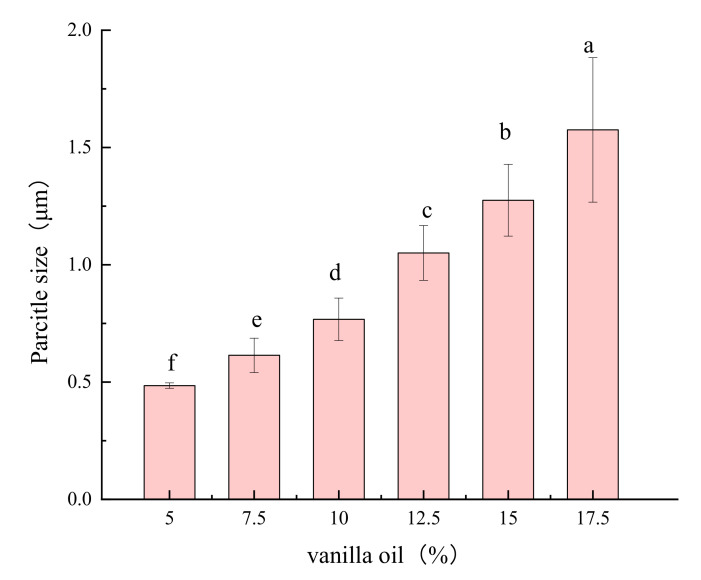
OSA−starch with different vanilla oil content prepared emulsion particle size. Note: Different letters represent significant differences (*p* < 0.05).

**Figure 6 foods-11-00987-f006:**
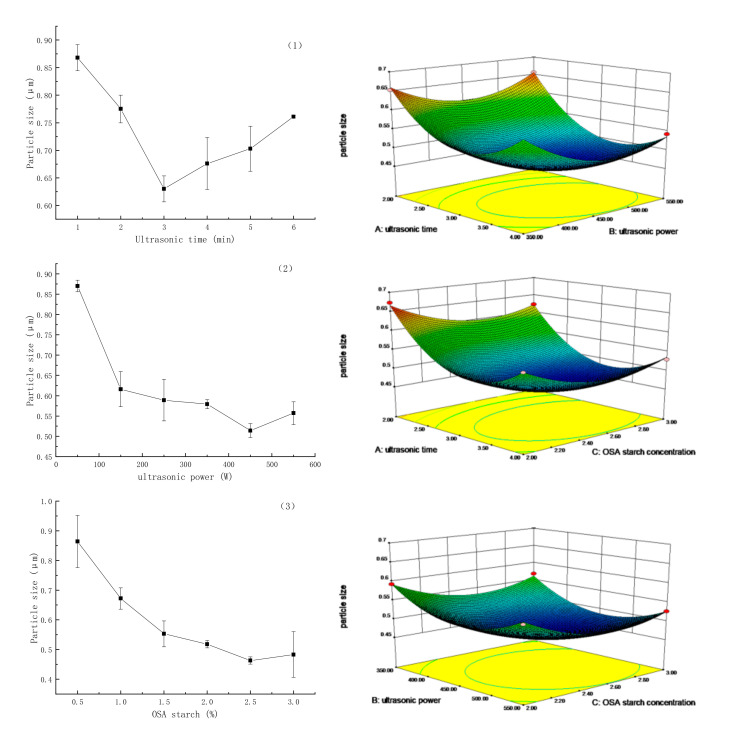
(**Left**) Effects of different factors on emulsion particle size. (**right**) Optimization of emulsion preparation process by response surface methodology.

**Figure 7 foods-11-00987-f007:**
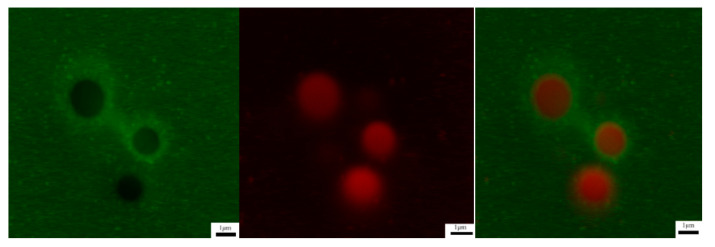
Confocal laser scanning microscopy images of Pickering emulsion of vanilla oil 600×.

**Figure 8 foods-11-00987-f008:**
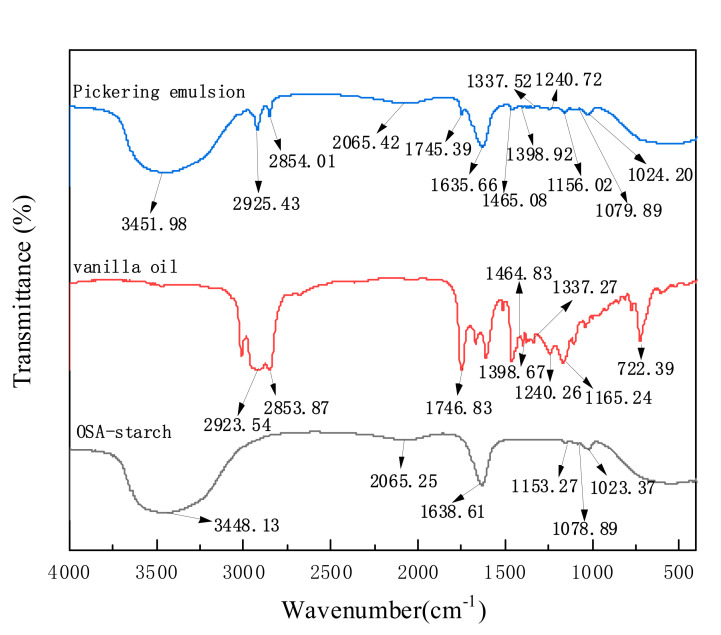
FT−IR spectra of OSA−starch solution, vanilla essential oil, and Pickering emulsion.

**Figure 9 foods-11-00987-f009:**
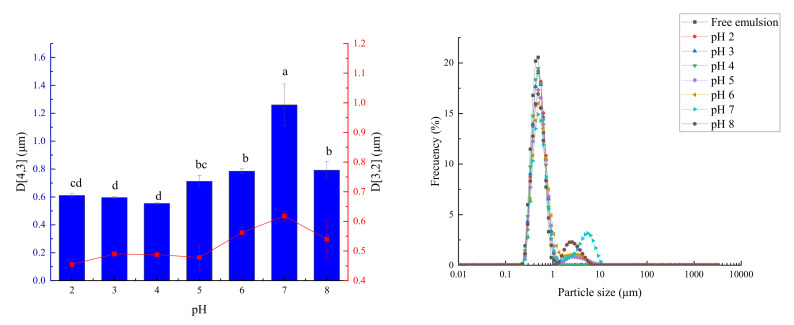
The change of emulsion particle size under different pH (**left**) and the particle size distribution diagram (**right**). Note: Different letters represent significant differences (*p* < 0.05).

**Figure 10 foods-11-00987-f010:**
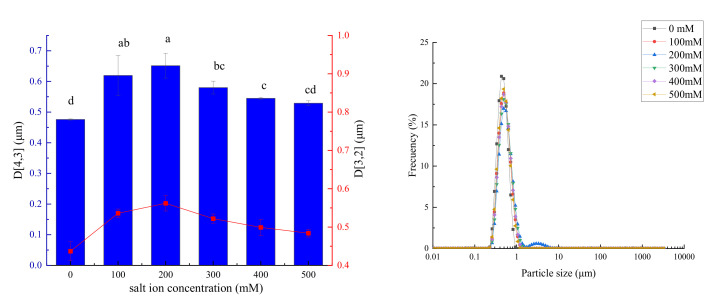
The change of emulsion particle size under different salt ion concentrations (**left**) and particle size distribution diagram (**right**). Note: Different letters represent significant differences (*p* < 0.05).

**Figure 11 foods-11-00987-f011:**
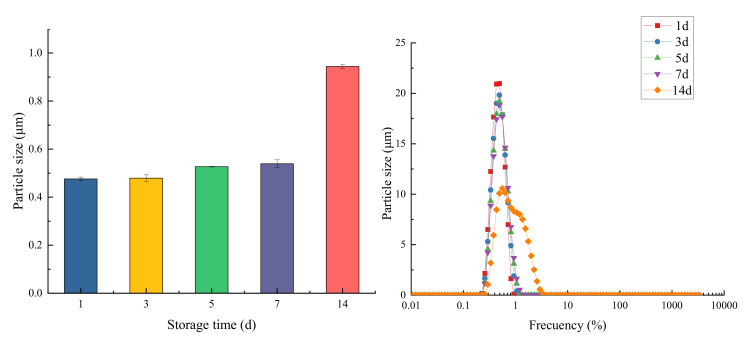
The change of emulsion particle size under different storage times (**left**) and its particle size distribution diagram (**right**).

**Figure 12 foods-11-00987-f012:**
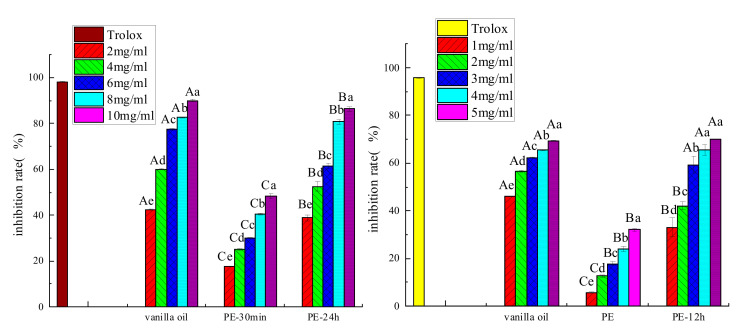
(**Left**) DPPH· scavenging ability and (**right**) ABTS**·****^+^** scavenging ability of vanilla oil and Pickering emulsion. Note: a–e represent the difference between different concentrations in the same group, A–C represent the difference between different groups at the same concentration, and different letters represent significant differences (*p* < 0.05).

**Table 1 foods-11-00987-t001:** The factors and levels of response surface test.

Level	OSA Concentration (%)	Ultrasonic Power (W)	Ultrasound Time (min)
−1	2	350	2
0	2.5	450	3
1	3	550	4

**Table 2 foods-11-00987-t002:** The chemical composition of vanilla essential oil.

No.	Compound	RI_(lit.)_	RI_(calc__.__)_	Retention Time	Formula	Relative Content (%)
1	Styrene	795	790	6.30	C_8_H_8_	0.23 ± 0.14 ^d^
2	2,2,4,6,6-pentamethyl-heptane,	980	981	9.72	C_12_H_26_	0.15 ± 0.11 ^d^
3	4-methoxy-benzaldehyde,	1244	1249	25.65	C_8_H_8_O_2_	0.14 ± 0.10 ^d^
4	4-methoxy-benzenemethanol,	1278	1283	28.25	C_8_H_10_O_2_	8.69 ± 2.13 ^b^
5	4-methoxy-benzoic acid methyl ester	1371	1368	34.78	C_9_H_10_O_3_	0.17 ± 0.12e ^d^
6	3-phenyl-2-Propenoic acid methyl ester (E)-	1380	1388	35.45	C_10_H_10_O_2_	0.03 ± 0.01 ^d^
7	Vanillin	1403	1421	37.53	C_8_H_8_O_3_	30.54 ± 1.59 ^a^
8	Caryophyllene	1492	1472	37.94	C_15_H_24_	0.10 ± 0.08 ^d^
9	4-methoxy-benzenemethanol acetate	1524	1533	38.48	C_10_H_12_O_3_	0.18 ± 0.11 ^d^
10	4-methoxy benzoic acid	1570	1575	41.84	C_8_H_8_O_3_	0.84 ± 0.08 ^d^
11	Dodecanoic acid	1586	1587	50.30	C_12_H_24_O_2_	0.06 ± 0.01 ^d^
12	Myristic acid	1770	1769	57.29	C_14_H_28_O_2_	0.08 ± 0.05 ^d^
13	2-Pentadecanone, 6,10,14-trimethyl-	1848	1840	58.60	C_18_H_36_O	0.12 ± 0.06 ^d^
14	Hexadecanoic acid, methyl ester	1927	1925	59.72	C_17_H_34_O_2_	0.12 ± 0.07 ^d^
15	n-Hexadecanoic acid	1961	1974	60.30	C_16_H_32_O_2_	4.83 ± 0.12 ^c^
16	Hexadecanoic acid, ethyl ester	1996	1986	60.54	C_18_H_36_O_2_	0.15 ± 0.07 ^d^
17	9,12-Octadecadienoic acid (Z, Z)-, methyl ester	2100	2089	61.77	C_19_H_34_O_2_	0.65 ± 0.26 ^d^
18	11-Octadecenoic acid, methyl ester	2101	2095	61.84	C_19_H_36_O_2_	0.17 ± 0.11 ^d^
19	Linoleic acid	2104	2106	62.47	C_18_H_32_O_2_	3.81 ± 2.59 ^c^
20	9,12-Octadecadienoic acid, ethyl ester	2134	2139	62.68	C_20_H_36_O_2_	1.16 ± 0.58 ^d^
21	Oleic Acid	2167	2161	62.76	C_18_H_34_O_2_	0.73 ± 0.45 ^d^
22	Heneicosane	2200	2199	63.13	C_21_H_44_	0.56 ± 0.06 ^d^
23	Tetracosane	2300	2305	64.20	C_24_H_50_	1.39 ± 0.98 ^d^
24	Nonadecane-2,4-dione	2400	2396	65.13	C_19_H_36_O_2_	1.06 ± 0.77 ^d^
25	1-Octacosanol	2456	2465	65.58	C_28_H_58_O	1.58 ± 0.56 ^d^
26	Octacosane	2600	2595	65.75	C_28_H_58_	4.31 ± 0.35 ^c^
27	Heptacosane	2700	2698	67.36	C_27_H_56_	3.43 ± 0.44 ^c^

Note: RI_(lit)_: The retention index reference value was determined by the database (www.odour.org.uk, accessed on 19 February 2022) and reported in the literature [1,2,3,4,5]. RI_(calc)_: Experimental mass spectrometry results. Means of triplicate measurements in the same column ± standard deviation with different letters are significantly different (*p* < 0.05).

**Table 3 foods-11-00987-t003:** Fatty acid composition of vanilla essential oil.

No.	Fatty Acid	Formula	Retention Time	Relative Content (%)
1	Myristic acid (C14:0)	C_14_H_28_O_2_	30.461	0.07 ± 0.04 ^e^
2	Palmitic acid (C16:0)	C_16_H_32_O_2_	36.069	19.91 ± 0.21 ^b^
3	Stearic acid (C18:0)	C_18_H_36_O_2_	42.662	7.88 ± 0.12 ^d^
4	Oleic acid (C18:1 n − 9)	C_18_H_34_O_2_	43.856	13.32 ± 1.62 ^c^
5	Linoleic acid (C18:2 n − 6)	C_18_H_32_O_2_	46.734	58.07 ± 3.49 ^a^
6	Linolenic acid (C18:3 n − 3)	C_18_H_30_O_2_	50.490	0.74 ± 0.28 ^e^

Means of triplicate measurements in the same column ± standard deviation with different letters are significantly different (*p* < 0.05).

**Table 4 foods-11-00987-t004:** Characteristic value of vanilla essential oil.

AV (mg KOH/g Oil)	PV (mg/g)	IV (mg/100 g Oil)	SN (mg KOH/g Oil)
0.68 ± 0.15	0.36 ± 0.02	59.22 ± 0.32	147.26 ± 1.58

**Table 5 foods-11-00987-t005:** Experimental points of the Box Behnken design and the experimental data.

Run No.	OSA Concentration (%)	Ultrasonic Power (W)	Ultrasound Time (min)	Particle Size D_4,3_ (μm)
1	0 (2.5)	1 (550)	1 (4)	0.487 ± 0.023 ^c^
2	1 (3)	1	0 (3)	0.529 ± 0.052 ^bc^
3	−1 (2)	0 (450)	−1 (2)	0.645 ± 0.156 ^abc^
4	−1	1	0	0.531 ± 0.112 ^bc^
5	1	−1 (350)	0	0.565 ± 0.056 ^bc^
6	0	0	0	0.566 ± 0.063 ^bc^
7	−1	−1	0	0.612 ± 0.058 ^abc^
8	0	0	0	0.560 ± 0.066 ^bc^
9	−1	0	1	0.526 ± 0.152 ^bc^
10	0	−1	−1	0.782 ± 0.231 ^a^
11	1	0	1	0.499 ± 0.059 ^c^
12	0	0	0	0.561 ± 0.032 ^bc^
13	0	−1	1	0.543 ± 0.045 ^bc^
14	0	1	−1	0.791 ± 0.011 ^a^
15	0	0	0	0.562 ± 0.044 ^bc^
16	0	0	0	0.563 ± 0.036 ^bc^
17	1	0	−1	0.693 ± 0.143 ^ab^

^a,b,c^ Means in same column with different letters are significantly different at *p* < 0.05.

## Data Availability

The data presented in this study are available on request from the corresponding author.

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
