# Peer review of "Octenyl Succinic Acid Starch-Stabilized Vanilla Essential Oil Pickering Emulsion: Preparation, Characterization, Antioxidant Activity, and Storage Stability"

_foods, 2022, doi:10.3390/foods11070987_

Round 1
Reviewer 1 Report
Lines 55-54 Essential oils are extracted through different techniques including steam and organic solvent extraction, supercritical CO2 extraction and so on. Please add more details instead so on.
Lines 100-101. Indicate the size of the crushed pods and how was the powder obtained (the equipment used) together with the size of the particles.
Line 106. Correct the typo in gas-chromatography title. Why the Kovats Index (LRI) was not calculated?
Lines 135-138. Please indicate how the temperature was controlled and the value of temperature during emulsification by US.
Lines 143-146. First indicate equipment then sample treatment. Please put in order the sentences.
Lines 155-161. During baking was there any volatilization of the aromatic compounds?
Lines 186-187 “All experiments were performed in triplicate to determine the statistical significance of the findings, and the results were recorded as means and standard deviations.” The statistical significance should not be mentioned next to mean and Stdev. Please remove it from the sentence.
Lines 201-203. “The volatile components of vanilla analyzed by GC-MS have been mainly focused on organic solvent extraction, while the composition of vanilla essential oil extracted by supercritical fluid has not been reported.” It is very difficult to understand why the GC-MS of SC-CO2 oil was not performed. Please provide a rationale in the text. At the same time in lines 212-214 the results are compared with ones obtained by SC-CO2 ( Yamini et al.). Please explain why.
From figure 2, it can be seen that after 60 min there is some bleeding, however an important peak at 66.75 RT has no correspondent in the RT indicated in Table 2. Please explain the differences in RT from the table compared to chromatogram.
Lines 218-228. Please clarify it is a higher percentage… .
Lines 221-222. The assumption is too speculative that a difference of less than 1áµ’C could be responsible for differences. Please reconsider.
Line 239. Write myristic with small caps
Line 249. ” The vanilla essential oil showed the lower saturated fatty acid……” Please delete the second the
Lines 261-262. Do not repeat the values from Table 4 in the text.
Line 284.”….starch particles to be better enclose the surface of the droplets.” Please rephrase. It is unclear.
Lines 351-363. Please correct font and equation written. Explain the factors A,B,C, E. What is (2.500E-003) ? Why D was not considered?
Line 417. “Although the particle size increased slightly, this change was not significant.” Please indicate p-value.
Figure 10. The image resolution of the particle size vs. storage time is not clear. Please improve image quality.
Lines 465-466. “..after 24, indicating that the scavenging ability of the Pickering emulsion was higher than that of pure vanilla essential oil and PE was released slowly”. From fig. 11 it doesn`t show a higher scavenging ability than the oil…it is just comparable. Please explain.
Author Response
Cover letter of the manuscript “Octenyl succinic acid starch-stabilized vanilla essential oil Pickering emulsion: preparation, characterization, antioxidant activity, and storage stability”
Answer to Reviewer: 1
Specific comments:
Question 1: Lines 55-54 Essential oils are extracted through different techniques including steam and organic solvent extraction, supercritical CO2 extraction and so on. Please add more details instead so on.
Answer: Thank you. The sentence “Essential oils are extracted through different techniques including steam and organic solvent extraction, supercritical CO2 extraction and so on.” has been changed “Essential oils are extracted through different techniques including steam distillation and organic solvent extraction, supercritical CO2 extraction, ultrasonic-assisted extraction, ultrasonic extraction and molecular distillation, respectively.” (Please see page 2, line 55-58, in the revised manuscript).
Question 2: Lines 100-101. Indicate the size of the crushed pods and how was the powder obtained (the equipment used) together with the size of the particles.
Answer: Thank you. The size of vanilla pods was about 7 cm. The pods were cut to 2-3 cm and then placed in a high speed disintegrator (Herbal Crusher YF-1000). The size of crushed pod powders was about 200 μm. The sentence “First, the fermented vanilla pods were crushed, placed in an oven to reduce the moisture content to 6%, and stored.” was changed to “First, the sample of fermented vanilla pods were dehydrated in electric blast drying oven at 50 ± 5.0â—¦C for 30 h to eliminate moisture water content of 6%. The size of vanilla pods was about 7 cm. The pods were cut to 2-3 cm and then placed in a high speed disintegrator (Herbal Crusher YF-1000). The size of crushed pod powders was about 200 μm”. (Please see page 3, line 105-111, in the revised manuscript).
Question 3: Line 106. Correct the typo in gas-chromatography title. Why the Kovats Index (LRI) was not calculated?
Answer: Thanks. The typo in gas-chromatography title “Gas chromat2ography-mass spectrometry conditions for vanilla essential oil and fatty acid” was changed to “Gas chromatography-mass spectrometry conditions for vanilla essential oil and fatty acid”. Kovats Index (LRI) was calculated supplemented in the Table 2 below. (Please see page 3, line 115; page 6, line 214 in the revised manuscript).
Table 2. The chemical composition of vanilla essential oil.
|
No. |
Compound |
RI (lit) |
RI (calc) |
Retention time |
Formula |
Relative content (%) |
|
1 |
Styrene |
795 |
790 |
6.30 |
C8H8 |
0.23±0.14d |
|
2 |
2,2,4,6,6-pentamethyl-heptane, |
980 |
981 |
9.72 |
C12H26 |
0.15±0.11d |
|
3 |
4-methoxy-benzaldehyde, |
1244 |
1249 |
25.65 |
C8H8O2 |
0.14±0.10d |
|
4 |
4-methoxy-benzenemethanol, |
1278 |
1283 |
28.25 |
C8H10O2 |
8.69±2.13b |
|
5 |
4-methoxy-benzoic acid methyl ester |
1371 |
1368 |
34.78 |
C9H10O3 |
0.17±0.12ed |
|
6 |
3-phenyl-2-Propenoic acid methyl ester (E)- |
1380 |
1388 |
35.45 |
C10H10O2 |
0.03±0.01d |
|
7 |
Vanillin |
1403 |
1421 |
37.53 |
C8H8O3 |
30.54±1.59a |
|
8 |
Caryophyllene |
1492 |
1472 |
37.94 |
C15H24 |
0.10±0.08d |
|
9 |
4-methoxy-benzenemethanol acetate |
1524 |
1533 |
38.48 |
C10H12O3 |
0.18±0.11d |
|
10 |
4-methoxy benzoic acid |
1570 |
1575 |
41.84 |
C8H8O3 |
0.84±0.08d |
|
11 |
Dodecanoic acid |
1586 |
1587 |
50.30 |
C12H24O2 |
0.06±0.01d |
|
12 |
Myristic acid |
1770 |
1769 |
57.29 |
C14H28O2 |
0.08±0.05d |
|
13 |
2-Pentadecanone, 6,10,14-trimethyl- |
1848 |
1840 |
58.60 |
C18H36O |
0.12±0.06d |
|
14 |
Hexadecanoic acid, methyl ester |
1927 |
1925 |
59.72 |
C17H34O2 |
0.12±0.07d |
|
15 |
n-Hexadecanoic acid |
1961 |
1974 |
60.30 |
C16H32O2 |
4.83±0.12c |
|
16 |
Hexadecanoic acid, ethyl ester |
1996 |
1986 |
60.54 |
C18H36O2 |
0.15±0.07d |
|
17 |
9,12-Octadecadienoic acid (Z, Z)-, methyl ester |
2100 |
2089 |
61.77 |
C19H34O2 |
0.65±0.26d |
|
18 |
11-Octadecenoic acid, methyl ester |
2101 |
2095 |
61.84 |
C19H36O2 |
0.17±0.11d |
|
19 |
Linoleic acid |
2104 |
2106 |
62.47 |
C18H32O2 |
3.81±2.59c |
|
20 |
9,12-Octadecadienoic acid, ethyl ester |
2134 |
2139 |
62.68 |
C20H36O2 |
1.16±0.58d |
|
21 |
Oleic Acid |
2167 |
2161 |
62.76 |
C18H34O2 |
0.73±0.45d |
|
22 |
Heneicosane |
2200 |
2199 |
63.13 |
C21H44 |
0.56±0.06d |
|
23 |
Tetracosane |
2300 |
2305 |
64.20 |
C24H50 |
1.39±0.98d |
|
24 |
Nonadecane-2,4-dione |
2400 |
2396 |
65.13 |
C19H36O2 |
1.06±0.77d |
|
25 |
1-Octacosanol |
2456 |
2465 |
65.58 |
C28H58O |
1.58±0.56d |
|
26 |
Octacosane |
2600 |
2595 |
65.75 |
C28H58 |
4.31±0.35c |
|
27 |
Heptacosane |
2700 |
2698 |
67.36 |
C27H56 |
3.43±0.44c |
Note: RI (lit): The retention index reference value was determined by the database (www.odour.org.uk) and reported in literature [1-5]. RI (calc): Experimental mass spectrometry results
Question 4: Lines 135-138. Please indicate how the temperature was controlled and the value of temperature during emulsification by US.
Answer: Thank you. During phacoemulsification, the temperature was broadly maintained at 25°C ± 3.0°C. In order to control the temperature of samples, the sample was placed in a water bath with a larger beaker. The ultrasonic time was set to 4s, and the interval time was set to 8s for facilitating heat dissipation. And this part has been supplemented to 2.5 Preparation of Pickering emulsion “Maintain the temperature at 25°C ± 3°C during ultrasonication.” (Please see page 3, line 139-141, in the revised manuscript).
Question 5: Lines 143-146. First indicate equipment then sample treatment. Please put in order the sentences.
Answer: Thanks a lot. The Mastersizer 3000 was used to determines the particle size of Pickering emulsions. The sentence “Pickering emulsion diluted 100 times with distilled water. Particle size distribution analysis was performed using a Mastersizer3000 at 25 ± 1°C.” was changed to “Mastersizer 3000 was used to measure the mean sizes and particle size distribution of the Pickering emulsion at 25 ± 1°C.” (Please see page 4, line 153-156, in the revised manuscript).
Question 6: Lines 155-161. During baking was there any volatilization of the aromatic compounds?
Answer: Thanks. No aromatic compounds were volatilized during baking. In order to avoid misunderstanding, the sentence “The sample was baked for 60 s in a rapid infrared dryer.” was changed to “The sample was kept for 60 s in a rapid infrared dryer.” (Please see page 4, line 169, in the revised manuscript).
Question 7: Lines 186-187 “All experiments were performed in triplicate to determine the statistical significance of the findings, and the results were recorded as means and standard deviations.” The statistical significance should not be mentioned next to mean and Stdev. Please remove it from the sentence.
Answer: Thanks. “All experiments were performed in triplicate to determine the statistical significance of the findings, and the results were recorded as means and standard deviations.” was changed to “All experiments were performed in triplicate, and the results were recorded as means and standard deviations.” (Please see page 5, line 202-203, in the revised manuscript).
Question 8: Lines 201-203. “The volatile components of vanilla analyzed by GC-MS have been mainly focused on organic solvent extraction, while the composition of vanilla essential oil extracted by supercritical fluid has not been reported.” It is very difficult to understand why the GC-MS of SC-CO2 oil was not performed. Please provide a rationale in the text. At the same time in lines 212-214 the results are compared with ones obtained by SC-CO2 (Yamini et al.). Please explain why.
Answer: Thanks a lot. Vanilla essential oil contains a lot of volatiles including aldehydes, acids, alcohols and non-volatiles such as ester of fatty acid, resins, pigments. Solvent-extracted vanilla essential oil contains a wider range of volatile constituents. Supercritical CO2 extraction conditions are difficult to control, and the extracted essential oil contains small molecules, macromolecular substances and resins. As a result. vanilla essential oil mostly adopts solvent extraction method instead of supercritical CO2 extraction at present. In our previous study, a large number of experiments of supercritical CO2 extraction of vanilla essential oil were conducted to determine the optimal extraction conditions. and then the analysis of vanilla essential oil was carried out by GC-MS component.
The supercritical CO2 extraction mentioned in the Yamini literature is Salvia mirzayanii essential oil instead of vanilla essential oil. It was showed that the supercritical CO2 extraction method can obtain fewer components but higher relative content compared with other extraction methods. In order to avoid misunderstanding, the sentence “Our present results were broadly similar with Yamini et al.” was changed to “Similar result was also observed of Yamini et al. who reported that the composition of the Salvia mirzayanii essential oil extracted by supercritical CO2” (Please see page 7, line 218-220, 229-232, in the revised manuscript).
Question 9: From figure 2, it can be seen that after 60 min there is some bleeding, however an important peak at 66.75 RT has no correspondent in the RT indicated in Table 2. Please explain the differences in RT from the table compared to chromatogram.
Answer: Thank you. As your suggested, the figure 3 has been corresponded to table 2. (Please see page 6, lines 214, in the revised manuscript).
Table 2. The chemical composition of vanilla essential oil.
|
No. |
Compound |
RI (lit) |
RI (calc) |
Retention time |
Formula |
Relative content (%) |
|
1 |
Styrene |
795 |
790 |
6.30 |
C8H8 |
0.23±0.14d |
|
2 |
2,2,4,6,6-pentamethyl-heptane, |
980 |
981 |
9.72 |
C12H26 |
0.15±0.11d |
|
3 |
4-methoxy-benzaldehyde, |
1244 |
1249 |
25.65 |
C8H8O2 |
0.14±0.10d |
|
4 |
4-methoxy-benzenemethanol, |
1278 |
1283 |
28.25 |
C8H10O2 |
8.69±2.13b |
|
5 |
4-methoxy-benzoic acid methyl ester |
1371 |
1368 |
34.78 |
C9H10O3 |
0.17±0.12ed |
|
6 |
3-phenyl-2-Propenoic acid methyl ester (E)- |
1380 |
1388 |
35.45 |
C10H10O2 |
0.03±0.01d |
|
7 |
Vanillin |
1403 |
1421 |
37.53 |
C8H8O3 |
30.54±1.59a |
|
8 |
Caryophyllene |
1492 |
1472 |
37.94 |
C15H24 |
0.10±0.08d |
|
9 |
4-methoxy-benzenemethanol acetate |
1524 |
1533 |
38.48 |
C10H12O3 |
0.18±0.11d |
|
10 |
4-methoxy benzoic acid |
1570 |
1575 |
41.84 |
C8H8O3 |
0.84±0.08d |
|
11 |
Dodecanoic acid |
1586 |
1587 |
50.30 |
C12H24O2 |
0.06±0.01d |
|
12 |
Myristic acid |
1770 |
1769 |
57.29 |
C14H28O2 |
0.08±0.05d |
|
13 |
2-Pentadecanone, 6,10,14-trimethyl- |
1848 |
1840 |
58.60 |
C18H36O |
0.12±0.06d |
|
14 |
Hexadecanoic acid, methyl ester |
1927 |
1925 |
59.72 |
C17H34O2 |
0.12±0.07d |
|
15 |
n-Hexadecanoic acid |
1961 |
1974 |
60.30 |
C16H32O2 |
4.83±0.12c |
|
16 |
Hexadecanoic acid, ethyl ester |
1996 |
1986 |
60.54 |
C18H36O2 |
0.15±0.07d |
|
17 |
9,12-Octadecadienoic acid (Z, Z)-, methyl ester |
2100 |
2089 |
61.77 |
C19H34O2 |
0.65±0.26d |
|
18 |
11-Octadecenoic acid, methyl ester |
2101 |
2095 |
61.84 |
C19H36O2 |
0.17±0.11d |
|
19 |
Linoleic acid |
2104 |
2106 |
62.47 |
C18H32O2 |
3.81±2.59c |
|
20 |
9,12-Octadecadienoic acid, ethyl ester |
2134 |
2139 |
62.68 |
C20H36O2 |
1.16±0.58d |
|
21 |
Oleic Acid |
2167 |
2161 |
62.76 |
C18H34O2 |
0.73±0.45d |
|
22 |
Heneicosane |
2200 |
2199 |
63.13 |
C21H44 |
0.56±0.06d |
|
23 |
Tetracosane |
2300 |
2305 |
64.20 |
C24H50 |
1.39±0.98d |
|
24 |
Nonadecane-2,4-dione |
2400 |
2396 |
65.13 |
C19H36O2 |
1.06±0.77d |
|
25 |
1-Octacosanol |
2456 |
2465 |
65.58 |
C28H58O |
1.58±0.56d |
|
26 |
Octacosane |
2600 |
2595 |
65.75 |
C28H58 |
4.31±0.35c |
|
27 |
Heptacosane |
2700 |
2698 |
67.36 |
C27H56 |
3.43±0.44c |
Note: RI (lit): The retention index reference value was determined by the database (www.odour.org.uk) and reported in literature [1-5]. RI (calc): Experimental mass spectrometry results
Question 10: Lines 218-228. Please clarify it is a higher percentage….
Answer: Thank you. In order to avoid misunderstanding, the sentence “These results were in line with the conclusions of Dong et al.” was changed to “Similar results were obtained by Dong et al.”. (Please see page 8, lines 239-244, in the revised manuscript).
Question 11: Lines 221-222. The assumption is too speculative that a difference of less than 1áµ’C could be responsible for differences. Please reconsider.
Answer: Thanks a lot. In order to avoid misunderstanding, the sentence “The distinct result between our experiment and previous report was attributed to that temperature in this study was set at 36°C (309.2 K) compared to the 310.5 K.” has been deleted. (Please see page 7, lines 236-238, in the revised manuscript).
Question 12: Line 239. Write myristic with small caps
Answer: Thank for your circumspection. The “Myristic” was changed to “myristic”. (Please see page 8, line 255, in the revised manuscript).
Question 13: Line 249.” The vanilla essential oil showed the lower saturated fatty acid……” Please delete the second the
Answer: Thanks. “The second…” have been deleted. (Please see page 9, line 265-266, in the revised manuscript).
Question 14: Lines 261-262. Do not repeat the values from Table 4 in the text.
Answer: Thanks a lot. The duplicates referred to in Table 4 in the manuscript have been removed. (Please see page 9, line 276, in the revised manuscript).
Question 15: Line 284.”….starch particles to be better enclose the surface of the droplets.” Please rephrase. It is unclear.
Answer: Thank you. the sentence “….starch particles to be better enclose the surface of the droplets.” was change to “….starch particles to be better enclose the surface of the oil droplets.”(Please see page 10, line 297-298, in the revised manuscript).
Question 16: Lines 351-363. Please correct font and equation written. Explain the factors A,B,C, E. What is (2.500E-003) ? Why D was not considered?
Answer: Thanks. The font and equation written was corrected, “” was changed to “”. Moreover, A, B and C were response surface equation contains three factors A (OSA concentration), B (ultrasonic power) and C (ultrasound time). “E” stands for exponent. 2.500E-003 equal 0.0025. D was the absence of factors. (Please see page 12, line 367-368, in the revised manuscript).
Question 17: Line 417. “Although the particle size increased slightly, this change was not significant.” Please indicate p-value.
Answer: Thanks a lot. p-value was supplemented as “Although the particle size increased slightly, this change was not significant (P>0.05).”. (Please see page 14, lines 422, 431-433, in the revised manuscript).
Question 18: Figure 10. The image resolution of the particle size vs. storage time is not clear. Please improve image quality.
Answer: Thank you. The quality of image resolution of the particle size vs. storage time have been replaced as your suggested (Please see page 15, lines 465-466, in the revised manuscript).
Question 19: Lines 465-466. “..after 24, indicating that the scavenging ability of the Pickering emulsion was higher than that of pure vanilla essential oil and PE was released slowly”. From fig. 11 it doesn`t show a higher scavenging ability than the oil…it is just comparable. Please explain.
Answer: Thank you. As you said, Pickering emulsion-24 h antioxidant properties were similar to vanilla essential oil. In fact, after 24 h, the antioxidant capacity of Pickering emulsion was higher than 30 min, which was similar to the clearing capacity of vanilla essential oil. In order to avoid misunderstanding, “..after 24, indicating that the scavenging ability of the Pickering emulsion was higher than that of pure vanilla essential oil and PE was released slowly”. was changed to “indicating that the scavenging ability was higher than that of Pickering emulsion-30 min. it was also indicated that Pickering emulsion was released slowly.” (Please see page 16, lines 483-486, in the revised manuscript).

Reviewer 2 Report
The manuscript present lots of typo , grammatical and other errors. Besides these form-related issues, the content must be condensed. Some experimental aspects could be described in a better way.
The description of the experimental design is also somewhat confusing to me. Therefore, I heavily suggest the authors to rewrite the paper in a more compact and clearer way.

Author Response
Cover letter of the manuscript “Octenyl succinic acid starch-stabilized vanilla essential oil Pickering emulsion: preparation, characterization, antioxidant activity, and storage stability”
Answer to Reviewer: 2
The manuscript present lots of typo, grammatical and other errors. Besides these form-related issues, the content must be condensed. Some experimental aspects could be described in a better way.
The description of the experimental design is also somewhat confusing to me. Therefore, I heavily suggest the authors to rewrite the paper in a more compact and clearer way.
Answer: Thank you. We sincerely accept your comments on this manuscript. Many errors have been corrected in this manuscript, all figures and tables have been revised based on your comments. (Please see page 6, figure 2; page 6, figure 3; page 7, table 2; page 8, figure 4; page 9, table 4; page 9, figure 5; page 11, figure 6; page 13, figure 8; page 15, figure 10,11; page 16, figure 12, in the revised manuscript)
To better demonstrate the experimental design, a new Figure 2 has been supplemented. Furthermore, all results and discussions of the manuscript have been summarized and condensed.
First, the forewords have been revised. (Please see page 2, line 43; page 2, lines 49-50; page 2 lines 55-59; page 2, line 63; page 2, line 85, lines 90-91, in the revised manuscript).
Then, the experimental design section has been revised. (Please see page 3, lines 105-110; page 3, line 112; page 3, line 115; page 3, line 124; page 3, line 132; page 3, lines 139-141; page 4, lines 153-156; page 4, line 169; page 5, line 175; page 5, line 187; page 5, line 196; page 6, lines 202-203, in the revised manuscript).
Furthermore, the test results section has been revised. (Please see page 6, line 213; page 6, lines 215-217; page 7, lines 218-220, 229-232, 236-238; page 8, lines 249-244, 253, 255; page 9, lines 265-266, 276-277, 292; page 10, lines 297-299, 305, 315; page 11, lines 228-229; page 12, lines 363-364, 367-368, 380, 387-388; page 13, lines 393-394, 400; page 14, lines 421-422, 431-433, 443; page 15, lines 454-455, 465-466; page 16, 483-486, 495-496, 504-505, 518; page 17, lines 525-527, in the revised manuscript).
Finally, the corresponding contents related to references have also been revised. (Please see page 7, line 216; page 7, line 230; page 7, line 237; page 8, lines 240,242 ; page 9, line 266; page 17, lines 551-552, in the revised manuscript).

Round 2
Reviewer 1 Report
Figure 4. Please add measurement unit of Oy axis in Figure 4.
Figure 5, Figure 9, Figure 10. Please explain in the legend what the small caps letters represent
Figures 11 and 12. Please add significant differences between groups.
Lines 355- 367. Correct font.
Author Response
Specific comments:
Question 1: Figure 4. Please add measurement unit of Oy axis in Figure 4.
Answer: Thank you. The measurement unit of Oy axis has been added in Figure 4. (See page 8, line 242, in the revised manuscript).
Question 2: Figure 5, Figure 9, Figure 10. Please explain in the legend what the small caps letters represent
Answer: Thank you. The sentence “Note: Different letters represent significant differences (p<0.05).” has been added below the Figure 5, Figure 9 and Figure 10. (See page 9, line 282; page 13, lines 419; page 14, lines445 , in the revised manuscript).
Figure 5. OSA-starch with different vanilla oil content prepared emulsion particle size
Note: Different letters represent significant differences (p<0.05).
Figure 9. The change of emulsion particle size under different pH (left) and the particle size distribution diagram (right)
Note: Different letters represent significant differences (p<0.05).
Figure 10. The change of emulsion particle size under different salt ion concentrations (left) and particle size distribution diagram (right)
Note: Different letters represent significant differences (p<0.05).
Question 3: Figures 11 and 12. Please add significant differences between groups.
Answer: Thanks. Significant differences between groups and significant differences between groups have been added to the text. (See page 15, lines 482-488, in the revised manuscript).
Figure 12. (left) DPPH· scavenging ability and (right) ABTS·+ scavenging ability of vanilla oil and Pickering emulsion
Note: a~e represent the difference between different concentrations in the same group, A~C represent the difference between different groups at the same concentration, and different letters represent significant differences (p<0.05).
Question 4: Lines 355- 367. Correct font.
Answer: Thanks. “The regression equation model can be established by statistical analysis of the 17 groups of experimental data: The quadratic equation model was significant (P < 0.05) and R2Adj = 0.9484, indicating that the experimental design was reliable and that the fitted quadratic regression equation was suitable. The particle size of the Pickering emulsion could be predicted using this model under different preparation conditions. The minimum theoretical value of the Pickering emulsion was 0.46 μm. The when the optimal process conditions were ultrasonic time of 3.12 min, ultrasonic power of 479.03 W, OSA-starch concentration of 2.61%, respectively. To verify the best emulsification process, the optimal conditions were adjusted as starch concentration of 3%, ultrasonic power of 450 W, and ultrasonic time of 3 min, respectively. The verification test was carried out, and the particle size of the emulsion was 0.47 ± 0.01 μm which was close to the predicted value of 0.46 μm, indicating that the vanilla essential oil emulsion process model was optimized by the response surface method.” was be changed “The regression equation model can be established by statistical analysis of the 17 groups of experimental data: Y=0.46–0.047A-0.015B-0.018C-0.014AB+0.00375AC-0.0025BC+0.092A^2+0.057B^2+0.043C^2. The quadratic equation model was significant (P < 0.05) and R2Adj = 0.9484, indicating that the experimental design was reliable and that the fitted quadratic regression equation was suitable. The particle size of the Pickering emulsion could be predicted using this model under different preparation conditions. The minimum theoretical value of the Pickering emulsion was 0.46 μm. The when the optimal process conditions were ultrasonic time of 3.12 min, ultrasonic power of 479.03 W, OSA-starch concentration of 2.61%, respectively. To verify the best emulsification process, the optimal conditions were adjusted as starch concentration of 3%, ultrasonic power of 450 W, and ultrasonic time of 3 min, respectively. The verification test was carried out, and the particle size of the emulsion was 0.47 ± 0.01 μm which was close to the predicted value of 0.46 μm, indicating that the vanilla essential oil emulsion process model was optimized by the response surface method.” (See page 11, lines 355-368, in the revised manuscript).
Author Response
Cover letter of the manuscript “Octenyl succinic acid starch-stabilized vanilla essential oil Pickering emulsion: preparation, characterization, antioxidant activity, and storage stability”
Answer to Reviewer: 2
Thank you very much for your attention and the referee’s evaluation and comments on our paper. We sincerely hope this manuscript will be finally acceptable to be published on Foods. Thank you very much for all your help.